



# Brief Communication: Monitoring slope acceleration and impending failure with very high spatial and temporal resolution space borne Synthetic Aperture Radars

Andrea Manconi[1,2], Yves Bühler[1,2], Andreas Stoffel[1,2], Johan Gaume[1,2,3], Qiaoping Zhang[4], Valentyn Tolpekin[4]

[1] WSL Institute for Snow and Avalanche Research SLF, Flüelastrasse 11, CH-7260 Davos Dorf, Switzerland
[2] Climate Change, Extremes and Natural Hazards in Alpine Regions Research Centre CERC
[3] Institute for Geotechnical Engineering, ETH Zürich, Switzerland
[4] ICEYE Oy, Maarintie 6, 02150 Espoo, Finland

*Correspondence to*: Andrea Manconi (andrea.manconi@slf.ch)

**Abstract.** We demonstrate how high spatial and temporal resolution spaceborne synthetic aperture radar (SAR) imagery can be applied to improve slope deformation monitoring. We process ICEYE data acquired over the Brienz/Brinzauls slope instability in the Swiss Alps, where a catastrophic failure occurred on June 15[th], 2023. The available images provided unprecedented viewing of the moving slope from satellite SAR, with revisit times ranging from less than 1 hour to a maximum of 4 days. We apply image correlation algorithms (i.e., pixel-offset analysis) on SAR backscattering to measure surface velocity before the failure event and compared the results against ground-based SAR data used for early-warning purposes. We also compare pre- and post-failure imagery to map areas invaded by debris and to compute volumetric changes associated with the down wasted materials, showing good agreement with digital surface models generated from photogrammetric drone flights. Our results demonstrate how weather independent, high resolution satellite SAR data can provide data in critical scenarios of slope deformation, suggesting that crucial information can be retrieved timely also in remote, poorly accessible regions where in-situ monitoring is not viable.

## 1 Introduction

The availability of spaceborne Synthetic Aperture Radar (SAR) sensors has considerably increased in recent years due to launch of numerous missions, a trend expected to continue. This scenario offers new options for the systematic use of SAR datasets in numerous applications related to natural hazards, providing timely information independently of light and of weather conditions (Ban et al., 2020; Biggs et al., 2022; Monterroso et al., 2020; Leinss et al., 2020; Twele et al., 2016; Wangchuk et al., 2022). For what concerns slope stability analyses, thanks to the ESA Copernicus Sentinel-1 mission the applications progressively migrated from classic surveys, in the form of local and/or regional scale processing efforts for



inventorying purpose and/or for back analyses of case studies, to systematic/operational monitoring solutions (Barra et al., 2017; Bianchini et al., 2021). Several limitations remain, however, especially when the surface displacements sharply accelerate and the probability of slope failure increases (Manconi, 2021). In such cases, rapid assessment is demanded for early

warning and/or forecast applications aimed at protecting people and assets. New generation of SAR missions promise to be a game changer due to the massive technological improvement introduced, reducing the cost of production and the size of satellites and sensors, thus allowing for the space missions based on fleets and thus providing a more frequent revisit (Łukosz et al., 2021; Sigmundsson et al., 2024). In this work, we show the results of a pilot study performed during the crisis associated with the acceleration of the Brienz/Brinzauls slope instability (hereinafter referred to as Brienz), located in the Swiss alps,

culminating on June 15, 2023 with the failure of 1.2 $Mm^3$ slope portion named "the Island" (Loew, 2023). The large landslide complex is composed of a series of old rockslide events, fall deposits, and granular flows, which probably started after the last deglaciation ~13k years ago (Figi et al., 2022). Surface velocities increased consistently in recent times, and an extensive geodetic monitoring and an early warning system has been developed to protect people and infrastructures. Early warning is based on the combination of GNSS, robotized total station, a permanent ground-based SAR, several time lapse cameras, and

a doppler radar aimed at alarming and promptly closing an important road connection in case of rockfalls (Schneider et al., 2023). This emergency scenario gained resonance in the media due to the successful evacuation of the population (ca. 100 people) on May 13[th], 2023, i.e. ca. 1 month before the failure event. The main goal of this work is to investigate the reliability of the information provided by high spatial and temporal resolution satellite SAR, as well as their potential use in operational near-real-time monitoring and early warning contexts.

**2 Dataset**

We benefit of the high-resolution SAR imagery acquired from the ICEYE constellation over Brienz, and of the exceptional availability of independent measurements and ground observations allowing for a thoughtful validation. ICEYE is a commercial operator offering spaceborne SAR imagery (X-Band, 9.65GHz, wavelength 3.1 cm) acquired with different acquisition modalities (ICEYE, 2024). The current ICEYE constellation is composed of more than 30 units and rapidly

increasing. The current configuration allows covering the Earth's surface every 6 hours, although imaging modes and/or the geometrical characteristics may differ. For our application in Brienz, we considered imagery in Spot HIGH mode, covering an area of 25 $km^2$. Among the four different views available, the ascending orbit, left looking (ASC_L), provides an optimal sight over Brienz and minimal zones of layover and shadow (dark/black areas in Fig. 1b) as well as a good revisit, with 16 images acquired in the period before the failure event (see Table A1 and Fig.1c, red filled triangles, average revisit time 1.6 days,

minimum ca. 25 minutes, maximum 4 days). The images present high variability in terms of incidence angles, ranging from ca. 17 to ca. 37 degrees (see also Table A1). We used the GAMMA software (Wegmüller and Werner, 1997) to initially process the Single Look Complex (SLC) ICEYE images, applying a multilooking of 2 samples in azimuth and 1 in range, resulting in a pixel resolution cell of ca. 0.42m × 0.33m, in range and azimuth directions, respectively. SAR images were then aligned





(co-registered) considering as reference the acquisition on May-22, 2023 (ID1, see Table A1) and benefitting of a digital

topographic surface model (DSM) openly provided by Swisstopo over entire Switzerland (swissSURFACE3D, 2024, ground

sampling distance (GSD) of 0.50 m).

(a) Brienz / Brinzauls

(b) Lantsch / Lenz    Brienz / Brinzauls    Tiefencastel    GB-SAR

RTC (dB) -20   5    Fly direction (Azimuth)    Look direction (Range)    500m

(c) ICEYE ASC_R   ICEYE ASC_L   ICEYE DESC_R   ICEYE DESC_L   S1 ASC_R   S1 DESC_R   Slope failure

inc. angle (deg)

May-20   May-24   May-28   Jun-01   Jun-05   Jun-09   Jun-13   Jun-17   2023

**Figure 1: (a) Picture of the Brienz slope acquired from the GB-SAR station (see location in panel 1b) after the failure occurred on Jun-15, 2023. (b) Reference SAR (May-22, 2023) backscattering image in dB, after radiometric terrain correction (Small et al., 2022).**



**Some points of interest are indicated for spatial orientation. (c) ICEYE Spot HIGH imagery acquired over Brienz with indication of orbit direction (ascending or descending) and sensor look direction (left or right).**

### 3 Pre-event: measuring accelerated slope deformation.

Pixel-offset (hereafter referred to as PO, also known as image correlation, feature tracking, or speckle tracking) is an established approach to measure residual misalignments of digital images. For what concerns SAR imagery, PO is an efficient alternative to radar interferometry to measure displacements when large spatial gradients occur due to landslide processes (Manconi et al., 2014). As the accuracy of the measurements obtained with PO is directly related to the GSD of the imagery (Provost et al., 2022), we expect to detect displacements that occurred between two subsequent acquisitions in the order of

some centimetres.  We adapted the PO method presented in Bickel et al., (2018) by first converting the SAR backscattering in an orientation image  (Dematteis and Giordan, 2021), and using a template window of 128×128 pixels, overlap 75%, and oversampling factor of 4 (see details of the PO parameters in Bickel et al., 2018). We computed PO in the range and azimuth directions for all possible pairs with images acquired between May-22 and Jun-15, 2023, converted to meters multiplying by the pixel size (0.42m × 0.33m, in range and azimuth, respectively), calculated the magnitude of the 2D surface displacement

vectors in SAR coordinates, and finally generated velocity maps in m/day. Results for selected snapshots are shown in Fig 2. The acceleration of the Island is clearly visible compared with the rest of slope. Velocities increased from about 0.25 m/day reported on May-28, to a maximum of 0.5 m/day on June-03, 2023 (Fig. 2a). Thereafter, the surface velocities severely increased, reaching ca. 1 m/day on Jun-11 and then ramping up to maximum values of 4 m/day on Jun-15, i.e., the day of the failure. We extracted the maximum PO velocities for all pairs and generated velocity time series to compare with the

measurements obtained from a Ground-Based (GB) SAR installed in Brienz for early warning purposes. Fig. 2b shows the comparison between three different approaches to generate the time series from the PO results and the GB-SAR measurements. The least square approach (red triangles in Fig. 2b) considers the PO velocities of pairs when the difference in viewing angle is lower than 1.5 degrees, which was found to be optimal to avoid excess noise in the PO results due to changes of the SAR scene, thus not associated to real displacements. The least-square solution has been obtained by applying Singular Value

Decomposition (SVD, (Golub and Reinsch, 1971)). The method used here is to some extent similar to what applied by (Casu et al., 2011), where the system of equations for the PO time series was built by constraining the spatial baselines of SAR acquisitions to decrease the impact of system noise. The results compare well with the GB-SAR velocities (green line in Fig. 2b); however, this time series can be generated only if all the acquisitions until the failure occurrence are available, thus suitable for back analysis purposes but not ideal in an operational scenario of progressive acceleration. Thus, we generated time series

using two additional methods: (i) sequential, i.e., we computed the PO velocities related to the first available subsequent image, thus without constraints on the view angles (Fig. 2b, blue circles) and (ii) selected pairs, i.e. using sequentially only pairs with similar view angles (within 1.5 degrees range as used for the least-square approach, see Fig 2b, black diamonds). Also in these cases, the results follow well the evolution of the velocities reported with GB-SAR, albeit with more variability compared with the least-square results. Fig. 1c shows the conversion of the velocity time series into their inverse. When velocity increases





exponentially, its inverse approaches progressively lower values, and the linear projection of the time series to zero is expected to provide an indication of the ultimate stages towards an impending slope failure (Fukuzono, 1985).

**Figure 2: (a) Selected results showing the 2D surface velocities over Brienz measured with pixel-offset (PO) applied to the ICEYE**
**data. Note that the colour scale is logarithmic to show the extreme differences in velocities occurred within few days a factor six**



**larger compared to the one showed in (a). (b) Time series of maximum surface velocities recorded with PO vs. velocities recorded with GB-SAR. (c) Inverse velocity plot to show failure forecasting potential of ICEYE PO results vs. GB-SAR.**

Despite different versions and ad-hoc adaptations developed over the years (Sharifi et al., 2024), the inverse velocity approach is widely used in operational scenarios to infer the time of slope failure and to manage early warning scenarios (Rose and Hungr, 2007; Loew et al., 2017; Leinauer et al., 2023). Our results show that the inverse velocity values retrieved from PO applied to the ICEYE images converge comparably to the GB-SAR values in the week before the failure event occurred at Brienz on Jun-15, 2015. This suggests that the increase in spatial and temporal resolution of SAR datasets acquired from space
can provide new solutions for continuous monitoring when in-situ datasets are not available, and potentially be used in operational early warning scenarios for the evaluation of the slope time-of-failure.

**4 Post-event: slope failure documentation and volume estimation.**

Optical and radar images are used in the aftermath of landslide events to document and characterize the slope failure in terms of area and assets affected by the failed materials and to study the runout dynamics (Guzzetti et al., 2012). In Fig. 3a and 3b,
we show pre- and post-event ICEYE SPOT HIGH images, allowing to visually identify areas affected by changes (erosion and deposition) as well as assets invaded by the debris materials (i.e., the cantonal road connecting Brienz to the village of Lantsch/Lenz, see Fig. 1). Methods for change detection analyses, such as for example the Structural Similarity Index (SSI (Wang et al., 2004)) or the log-ratio (Mondini, 2017) applied to the SAR backscattering (shown Fig. 3c and 3d, respectively) can be used efficiently to map landslide boundaries and highlight important features associated to the failure dynamic, by
benefitting of the high spatial resolution of the herein used SAR images (Mondini et al., 2021). However, change detection provides only a 2-dimensional view of the event characteristics. To estimate the volume of the failed material, photogrammetric approaches based on space- or airborne optical imagery, as well as terrestrial and airborne LiDAR, are generally used, as they provide great flexibility and high-resolution results on the reconstruction of surface topography.

Spaceborne SAR can be also used for identification of topographic changes, provided that SAR scenes are acquired with the
same orbit and view angles, and that perpendicular baselines (i.e., orbit separation) between images acquired before and after the event remain within critical values for the determination of heights with interferometry (Bürgmann et al., 2000). Among the dataset we considered, two ICEYE images acquired after the Jun-15 event in Brienz (i.e. on Jun-18 and Jun-22, 2023, ascending orbit, right look) have same view angle and a spatial baseline of 931m resulting in a height of ambiguity of 7.6m. The differences between the topography before the failure event (swissSURFACE3D, 2024) and the digital surface models
obtained with radar interferometry (InSAR) are shown in Fig.4a. For validation, we compare the InSAR results with a DEM-of-Differences (DoD) obtained after photogrammetric drone flights before (Jun-8, 2023) and right after (Jun-16, 2023) the event (Fig. 4b). The georeferencing accuracy achieved with Post Processing Kinematic (PPK) is in the range of 0.1 m. The spatial distribution of the negative (erosion) and positive changes (deposition) due to the slope failure are well comparable.



However, the estimated negative volumes from InSAR are -0.79 Mm$^3$ vs. -1.24 Mm$^3$ from photogrammetry, i.e. a difference
of about 40%.

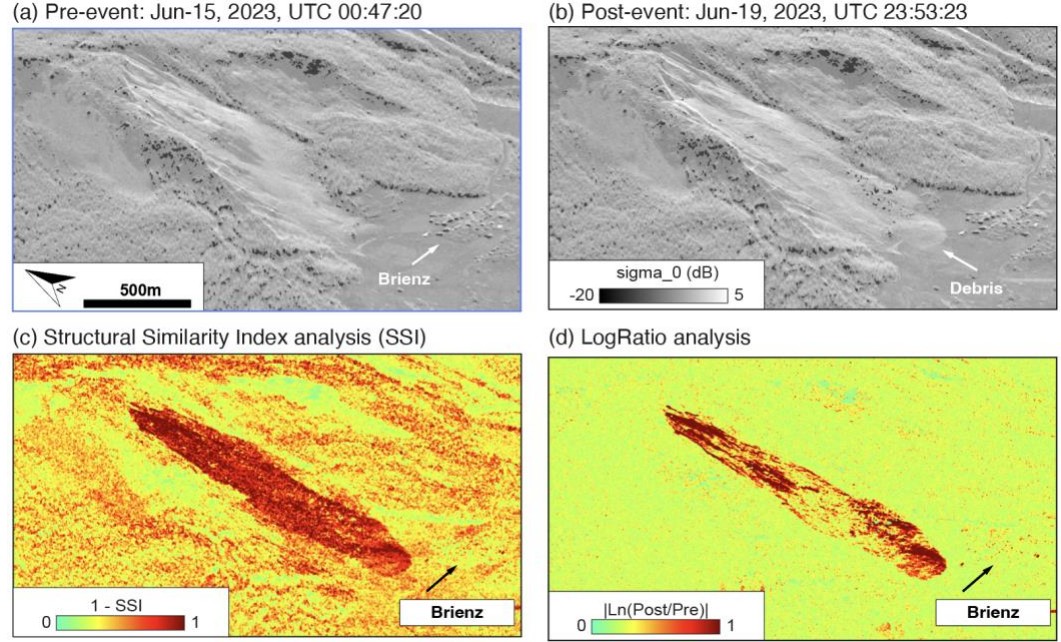

**Figure 3: Comparison between (a) pre- and (b) post- event ICEYE images acquired over Brienz. (c) and (d) show the results of analytical change detection attempts with SSI and log-ratio methods, respectively.**

These deviations in the source area are most likely due to phase unwrapping errors (procedure to convert the phase differences retrieved with InSAR into metric scale), associated to the presence of steep slopes and decorrelation due to large phase gradients. Another source of inaccuracy is atmospheric disturbance, as well as surface changes (minor slope failures) that occurred in the upper ranges of the slope after the main event. On the other end, deposited volumes from InSAR are +1.18 Mm$^3$, very similar to the +1.28 Mm$^3$ estimated from photogrammetry, only 10% difference.

## 5 Conclusions and Outlook

Reliable warning systems with high temporal and spatial resolution are the key to prevent life and asset loss caused by slope failure. In Brienz, an exceptional in-situ network allowed to recognize in due time the evolution towards a potential catastrophic failure, supporting the decision of local authorities to evacuate the population and restricting access to areas and infrastructures
at risk. However, such an extensive monitoring system is not replicable in all mountain areas where it would be needed, mainly due to logistic/access reasoning and/or financial matters. Satellite data, available in all weather conditions with sub-metric spatial resolutions and sub-daily revisit, can be a viable alternative to manage slope instability operational scenarios. As





demonstrated in this work, the results obtained from high resolution SAR images are well comparable with the ground-based

monitoring data and could be used also for near-real-time forecasting of time-of-failure with similar outcomes. In addition, the

165 increase in spatial and temporal resolution allows for prompt and accurate post-event mapping of the slope failure and could

support the determination of event magnitudes (volumes) via InSAR. Intrinsic limitations due to the combination of satellite

viewing geometries and slope orientation might still hinder the use of such an approach at high elevations. Moreover, in some

cases the unavailability of recent, high resolution topographic models might affect the accuracy of geocoding and decrease the

quality of the results. Despite, we have shown that the very high spatial resolution allows working efficiently also in SAR

coordinates, as the targets of interest and changes can be clearly identified. Our results provide a step forward towards the use

of satellite SAR imagery for operational scenarios, bringing new insights on the use of satellite SAR to monitor and potentially

forecast imminent slope failure in operational early warning scenarios. Accurate information provided timely by SAR satellites

day and night and in all-weather can be used in data-driven forecast methods as the inverse velocity approach shown here, as

well as for the calibration of more sophisticated numerical models aimed at the evaluation of slope stability and/or to simulate

runout scenarios (Bartelt et al., 2018; Gaume et al., 2024).

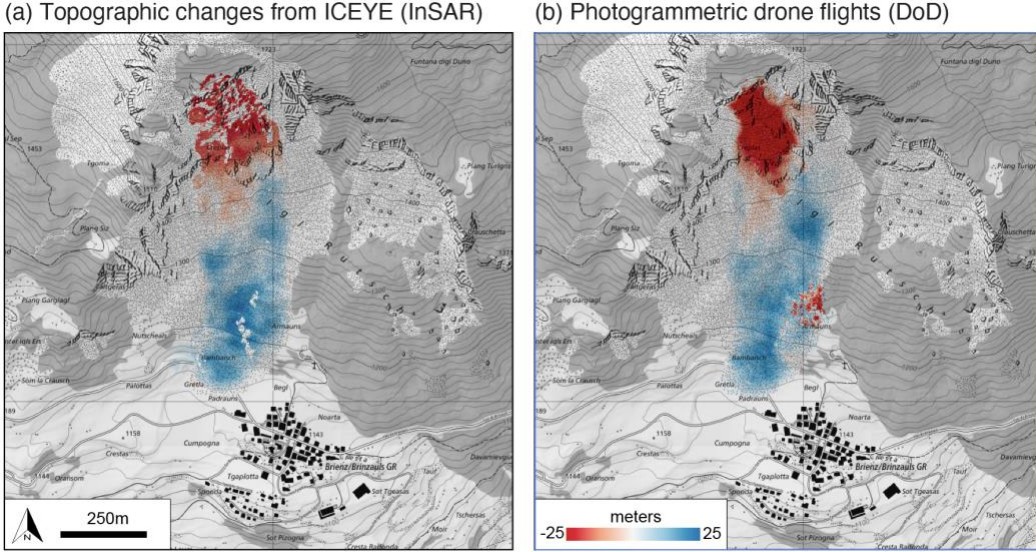

**Figure 4: Comparison between (a) ICEYE interferometry and (b) DSM-of-Difference (DoD) generated from photogrammetric drone flights. Note that the negative height difference in (b) in the lower part above the village are trees that were cut down by the**
180 **landslides.**



**Appendix**

| ID | Date & Time (UTC) | Time difference (days) | Inc. angle (degrees) |
|---|---|---|---|
| 1 | 2023.05.22 00:50:15 | - | 29.70 |
| 2 | 2023.05.23 20:12:39 | 1.8 | 34.05 |
| 3 | 2023.05.24 20:14:12 | 1.0 | 31.70 |
| 4 | 2023.05.28 00:50:34 | 3.19 | 29.90 |
| 5 | 2023.06.01 00:47:47 | 3.99 | 31.70 |
| 6 | 2023.06.03 00:50:12 | 2.0 | 31.17 |
| 7 | 2023.06.03 01:12:45 | 0.01 | 19.90 |
| 8 | 2023.06.06 19:51:52 | 3.77 | 24.70 |
| 9 | 2023.06.07 00:51:21 | 0.2 | 26.12 |
| 10 | 2023.06.07 01:06:24 | 0.01 | 36.70 |
| 11 | 2023.06.11 01:11:01 | 4.0 | 29.65 |
| 12 | 2023.06.11 19:53:55 | 0.77 | 20.50 |
| 13 | 2023.06.13 00:54:14 | 1.2 | 21.45 |
| 14 | 2023.06.14 01:03:47 | 1.0 | 37.20 |
| 15* | 2023.06.15 00:47:20 | 0.98 | 37.10 |
| 16 | 2023.06.15 01:15:19 | 0.02 | 22.27 |
| 17 | 2023.06.16 19:55:26 | 1.77 | 17.23 |
| 18** | 2023.06.18 22:55:43 | 2.12 | 43.5 |
| 19* | 2023.06.19 23:53:23 | 1.04 | 35.37 |
| 20** | 2023.06.22 22:13:03 | 2.93 | 43.5 |

**Table A1: List of ICEYE imagery (mode SPOT high, Ascending orbit, left looking sensor), date and time of acquisition (format (yyyy.mm.dd HH:MM:SS), relative temporal baselines (in days), and incidence angles. *Images used for detecting change between pre- and post- slope failure, selected due to similarity in incidence angle (see Fig. 3). **Images used for the estimation of topographic changes with radar interferometry (see Fig. 4).**

**Data and software availability**

The raw ICEYE Spot images used in this study can be downloaded from the ICEYE public archive (link will be included in the published version of the article). The GAMMA software used for the processing of the SAR images is available at



https://www.gamma-rs.ch/software. The PO algorithm used in this work is available at https://github.com/bickelmps/DIC_FFT_ETHZ.

## Author Contributions

AM conceived the study, processed the ICEYE imagery to generate the PO time series and the change detection analyses, produced the graphics, and wrote the manuscript. YB and AS acquired and processed the photogrammetric drone imagery to compute the volumetric changes shown in Figure 4 and revised the manuscript. JG provided funding resources for the acquisition of the ICEYE images and revised the manuscript. VT and QZ supported the acquisition of the ICEYE images, derived the InSAR results and revised the manuscript.

## Competing interests

At least one of the (co-)authors is a member of the editorial board of Natural Hazards and Earth System Sciences.

## Disclaimer

(will be included in the published version of the article)

## Acknowledgements

We thank Melanie Rankl and Michael Wollersheim from ICEYE Oy for supporting the data acquisition and fruitful discussions.

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
