# Peer review of "Brief Communication: Monitoring slope acceleration and impending failure with very high spatial and temporal resolution space borne Synthetic Aperture Radars"

_EGUsphere, 2024_

## Author Response (AR1)

Dear Editor

Thanks for handling our manuscript submission. We have now uploaded the revised version, and modified according to the reviewer's comments. Please find below the updated replies to the reviewers, for your reference. We hope this version improved the readibility and the quality of the text and graphical information provided, and we look forward receiving the final acceptance for publicationin NHESS.

Yours sincerely

Andrea Manconi

(on behalf of co-authors)

--

Reviewer #1

Manconi et al present a case study where they used high-resolution SAR images (ICEYE) to survey a fast moving landslide in the last few weeks before its failure. They measured the surface velocity and the volume of the collapsed material. Although they did not adopt innovative processing techniques, their aim is to show the potentialities of ICEYE in monitoring strongly dynamical processes, which I think is an interesting topic.

**Our reply: We thank Reviewer #1 for the positive evaluation of our manuscript**

Overall comments:

The manuscript is well written and the figures are clear. The motivation of the study is clear and the results support the thesis of the authors (i.e., the potentialities of ICEYE). However, I have two major concerns, pertaining to formal and methodological aspects. First, the manuscript is not compliant with the rules of brief communications: the abstract is too long (>180 words, but maximum 100 are admitted). There are four figures and one table (max three in total admitted). Probably figs. 3 and 4 might be merged and the table moved into supplementary material. Also the number of references (>30) is greater than the maximum admitted number (20). Since it is a brief communication, which is expected to deal with cutting edge research, I think that non recent references can be omitted. For example, I count at least 9 reference older that 10 years. There are six references at lines 29-30 in the introduction related to the use of SAR in natural

hazards, but probably one-two are sufficient. Plus, I have some doubts that the manuscript will be shorter than 4 pages in its final form, which is the maximum number for brief communications. If necessary, I think that the parts describing the time-series inversion and the DoD might be slightly shortened. If, on the contrary, the authors decide to convert the manuscript into a research article, I think that they should add more details to their study, for example a comparison with the results obtained with other satellites, but I do not suggest this option, since this work is already relevant in this version.

**Our reply: We have adapted the revised version to be compliant with Brief Communications standards.**

The second concern pertains to the lack of an uncertainty analysis, which is fundamental considering that the main goal of the manuscript is to demonstrate the ability of ICEYE to detect fast movements. Even the comparison with the GBSAR is presented only in a qualitative fashion, without any quantitative metrics.

**Our reply: Thanks for this comment. We added RMSE estimates (in the text and as Table in the appendix) of the difference between GB-SAR and offsets retrieved with the ICEYE time series.**

Besides these points, I have some other minor remarks:

- Title: I would add the mention to ICEYE in the title (or in the keywords, which I cannot see).

**Our reply: We modified the title in: "Brief Communication: Monitoring impending slope failure with very high-resolution space borne Synthetic Aperture Radars." We believe that the main message of the paper should be more genral and not directly associated with one or another data provider. The use of ICEYE is then mentioned in the abstract.**

- line 44: Even though in the literature the term robotized total station is occasionally adopted, I think that robotic is more correct English

**Our reply: modified**

- line 55: Can you explain in a few words the main characteristics of HIGH mode? E.g., terrain corrected or not, polarisation, etc,

**Our reply: We added details on the Spot mode. We have now modified according to the definitions provided by ICEYE and referred at https://www.iceye.com/sar-data/imaging-modes .**

- line 63: here and in the rest of the manuscript, a space is missing between numbers and units

**Our reply: modified.**

- lines 65-66: it is not clear how you benifitted from the DSM. Later in the manuscript you state that you used the SLC, thus I suppose that you did not orthorectify the SAR images. Please, can you exaplain this point better?

**Our reply: The DSM has been used for the co-registration of the SLC imagery and the computation of radiometric terrain corrected backscattering values. This calibration ensures a more appropriate evaluation of backscattering values and of their changes over space and time. We now cite the paper Small et al., 2022 where the procedure is explained.**

- line 76: PO is adopted also to measure surface displacement (like you do in this study), which, in my opinion, is a different concept compared to the residual alignment between images

**Our reply: we modified the sentence to avoid confusion.**

- line 79-80: You probably should exaplain that PO allows to detect sub-pixel displacement because non-expert readers might not figure out this statement. Besides, "some centimeter" sounds a little undefined

**Our reply: We modified and rendered the explanation clearer also for non-experts.**

- line 82 (factor of 4): This implies that the minimum measured displacement is 1/4 of the GSD, thus 8-10 cm. Peraphs you could move here the statement at lines 79-80.

**Our reply: done.**

- line 84 (pixel size): You adopt different terms to indicate this quantity: pixel resolution, GSD and pixel size. I suggest to be more uniform for clarity

**Our reply: done.**

- line 92: This paragraphs might be hard to understand to non-expert readers. First, the least square is not an approach to extract time-series, but the method adopted to solve the equation system. I suppose you are using a temporal closure-like method. In that case, you probably could cite Charrier et al (2022) and/or Hadrhi et al (2019), if relevant and if you can add references without exceeding 20 references, or move Casu et al (2011) here. For the sake of reproducibility, you could also specify whether you adopted weights and/or regularisation terms in the equation system.

**Our reply: We modified the paragraph. Interested readers are referred to Casu et al., 2011 for the explanation of the least square strategy used.**

- line 93: I think you should add some uncertainty analysis to your manuscript. For example, you could detect displacements of 10 cm, but what is the estimated uncertainty and how did you evaluate it? You could also show how the uncertainty is related to the relative viewing angles between the images. How did you determine the threshold of 1.5 degrees? Besides, the time-series inversion also introduces some uncertainty. Plus, you should add error bars in the time series plots.

**Our reply: done. We estimated the statistics in an area of 1000 x 1000 pixels considered stable. This provides the errorbars included in the time series plots and gives a quantitative estimate of the inaccuracies of the PO velocity. In most cases, they are below 10 cm**

- line 95: this is clear example of a reference that you can omit, since the SVD is a standard statistical technique.

**Our reply: removed.**

- lines 95-97: To what extend is your method similar to that of Casu et al (2011)? Since you did not provide any further detail, probably this statement can be omitted, leaving only the reference if necessary.

**Our reply: we adapted the sentence accordingly.**

- lines 98-99: I do not agree with this statement. The time-series inversion can be applied at any time. Of course, when new images become available, the results of the inversion might change, but, in a theoretically operative situation, one could calculate the time series with the currently available images and then update the time-series at every new acquisition.

**Our reply: We modified accordinlgly.**

- line 101 (fig2b): I do not see a direct correspondence between the blue circle in fig 2b and the available images in fig 1c. Probably, you did not consider pairs of images with a temporal baseline lower that a given threshold (1 day?) Plus, it seems that you have plotted the markers of the velocity in correspondence of the slave image, but I think that it is more correct to plot them in the middle time, since it represents the average velocity over the period between the two acquisitions.

**Our reply: Thanks for this comment. We have modified accordingly.**

- line 102 (black diamonds): In fig 2b I see more black diamonds that blue circles, but they should be less, since you only used a subset of the available images. Maybe the two markers are inverted? In any case, I would expect that one marker of the sequential approach would be present for each marker of the selected approach, but there are dates where only blue or black markers are shown.

**Our reply: Thanks for this comment. The "selected" PO time series (black dimaonds) consider all possible pairs with a constraint of 1.5 deg incidence angle, while the "sequential" (blue circles) consider no constrain on the angles but at least temporal baseline of 24 h, for this reason the diamonds are more than the blue circles. We hope the new text and the adapted figure will help clarifying this point.**

- line 103: How much more variability? You should provide some statistical metrics to compare the data and add an uncertainty analysis.

**Our reply: Done**

- fig 2c: I do not find a precise correspondence between the text and this figure (see comments above). Another concern pertains to the results of the time-series inversion (red triangles). In theory, there should be one result for each adopted image, but, on some dates, there are red triangles but not other markers and vice versa.

**Our reply: Thanks for this comment. We modified accordingly.**

- lines 123-124: since you have to shorten the manuscript, probably this is one statement that can be omitted.

**Our reply: the text was shortened and modified accordingly**

- line 169: Despite that

**Our reply: modified, thanks.**

Reviewer #2

Dear Authors,

Considering it as a brief communication, the paper is well-written and presents very interesting results on the potential of the new VHR SAR constellation ICEYE. I believe that the paper, as it stands, is suitable for publication.

**Our reply: We thank Reviewer #2 for the positive evaluation of our manuscript**

However, I think it would benefit from:

- Including a discussion of the look angle limits for the application of the PO method. It is not clear to me if ICEYE can precisely maintain the look angle over an area. This could be a constraint for the proposed approach.

**Our reply: Thanks for this comment. We have clarified in the revised version.**

- A more in-depth analysis of the interferometric DSM quality. This type of product is crucial and can be applied to a wide range of applications. In this context, it would be beneficial to include a more detailed analysis of the differences compared to UAV data.

**Our reply: Thanks for this comment. We computed the differences between the two DSMs and provided quantittive values.**

I hope this helps to improve.